

# Tracking precipitation features and associated large-scale environments over southeastern Texas

Ye Liu[1], Yun Qian[1], Larry K. Berg[1], Zhe Feng[1], Jianfeng Li[1], Jingyi Chen[1], and Zhao Yang[1]

[1]Pacific Northwest National Laboratory, Richland, WA 99352

*Correspondence to*: Ye Liu (ye.liu@pnnl.gov) and Yun Qian (yun.qian@pnnl.gov)

**Abstract**. Deep convection initiated under different large-scale environments exhibits different precipitation features and interacts with local meteorology and surface properties in distinct ways. Here, we analyze the characteristics and spatiotemporal patterns of different types of convective systems over southeastern Texas using 13 years of high-resolution observations and reanalysis data. We find that mesoscale convective systems (MCSs) contribute significantly to both mean
and extreme precipitation in all seasons, while isolated deep convection (IDC) plays a role in intense precipitation during summer and fall. Using self-organizing maps (SOMs), we found that convection can occur under unfavorable conditions without large-scale lifting or moisture convergence. In spring, fall and winter, frontal-related large-scale meteorological patterns (LSMPs) characterized by baroclinic waves and low-level moisture convergence act as primary triggers for convection, while the remaining storms are associated with an anticyclone pattern and orographic lifting. In summer, IDC are mainly
associated with front-related and anticyclones LSMPs, while MCSs occur more in frontal-related LSMPs. We further tracked the lifecycle of MCSs and IDC using the Flexible Object Tracker algorithm over southeastern Texas. MCSs frequently initiate west of Houston, travelling eastward for around 8 hours to southeastern Texas, while IDC initiate locally. The average duration of MCSs in southeastern Texas is 6.1 hours, approximately 4.1 times the duration of IDC. Diurnally, the initiation of convection associated with favorable LSMPs peak at 1100 UTC, 3 hours earlier than those associated with anticyclones.

**1 Introduction**

Deep convection is a major contributor to annual total precipitation and a source of very high intensity rainfall over coastal Texas (Feng et al., 2021; Li et al., 2021a; Houze, 2004). Deep convection can form as isolated deep convection (IDC) or grow into mesoscale convective systems (MCSs) under favorable thermodynamic and dynamic environments. The IDC are typical convective storms on the order to 10s of kilometers in horizontal scale and MCSs are convective systems with a contiguous
precipitation feature area on the order of 100 km or more in horizontal scale (Houze, 2004). Generally, MCSs have a longer duration, broader spatial coverage, and stronger precipitation. Rowe et al. (2012) reported that MCSs produce on average approximately 3 times the total precipitation amount than IDC over northwestern Mexico. MCS mean precipitation intensity is over seven times higher than the precipitation intensity of non-MCS events during the warm season over the central U.S.





(Hu et al. 2021). Moreover, strong seasonal variations have been found in both frequency and intensity of convective precipitation due to the varying large-scale and local environment (Rowe et al., 2012; Feng et al., 2019).

Located in southeastern Texas, Houston is the fifth most populous metropolitan area in the U.S. The city has been frequently threatened by severe floods and hail that are associated with convective storms (e.g., Brody et al., 2018; Collins et al., 2018; Nielsen and Schumacher, 2019; Valle-Levinson et al., 2020; Pryor et al., 2023). The predominant synoptic processes that affect convection initiation over southeastern Texas include the Bermuda high and Great Plains low-level jet (GPLLJ)

(Zhu and Liang, 2013; Wang et al., 2016; Whiteman et al., 1997). The Bermuda high is a semi-permanent high-pressure system forming over the Atlantic Ocean often in late spring. The GPLLJ refers to the climatological southerly wind that brings moisture from the Gulf of Mexico to the Great Plains, featuring a maximum wind speed in the lowest 1 km of the atmosphere (Whiteman et al., 1997; Berg et al., 2015; Yang et al., 2020). On one hand, the GPLLJ can be enhanced by the strong pressure gradient created by the west flank of the Bermuda high during its westward extension (Hodges and Pu, 2019; Wimhurst and

Greene, 2019), which is favorable for convection development. On the other hand, the westward extension of Bermuda high can create subsidence over the southeastern Texas, suppressing convection (Small and De Szoeke, 2007).

To understand the role of large-scale and local factors that impact convection initiation and development, Wang et al. (2022) isolated the influence of large-scale meteorology from microphysical and mesoscale influences, with the focus on summer climate over the southeastern Texas. They identified four large-scale meteorological patterns (LSMPs) characterized by the

location and strength of Bermuda High and GPLLJ, which are associated with convection initiation. Both the Bermuda High and GPLLJ are seasonal-varying systems, which exert different impacts on different type of convection. In this study, we extend the analysis from summer to other seasons and separate LSMPs associated with MCSs and IDC.

The rest of the paper is organized as follows: Section 2 describes the dataset and methodology, including the application of a convection dataset; Section 3 presents the results on the features of different types of convection over southeastern Texas,

linking convection to corresponding LSMP, and tracking of convective systems, as well as an analysis that focuses on the Houston metropolitan area to study fine-scale characteristics; Section 4 summaries the conclusions.

## 2 Data and Methods

This study utilizes a suite of existing datasets to investigate the nature of convective systems over southeast Texas, including a convection dataset and NEXRAD to characterize convective systems and a reanalysis dataset to identify LSMPs.

### 2.1 Convection dataset over the U.S.

The convection dataset used in this study is a high-resolution (4 km, hourly) observational product that covers the U.S. east of the Rocky Mountains from 2004 to 2017. This dataset contains the detailed classification, tracking, and characteristics of MCS and IDC events (Li et al., 2021a), available at doi: 10.25584/1632005. This dataset is developed by utilizing the Storm Labeling in Three Dimensions (SL3D) algorithm (Starzec et al., 2017) and an updated Flexible Object Tracker (Feng et al.,



2023; Li et al., 2021a) algorithm based on various datasets, including the National Centers for Environmental Prediction
(NCEP)/the Climate Prediction Center (CPC) L3 4 km Global Merged IR V1 brightness temperature dataset (Janowiak et al.,
2017), the three-dimensional (3D) Gridded Next-Generation Radar (NEXRAD) dataset (Bowman and Homeyer 2017), and
NCEP Stage IV precipitation dataset (Du, 2011). Cold cloud systems (CCSs) are first identified at each hour by searching for
cold cloud cores (regions with cloud base temperature $T_b$<225 K). The cold cloud cores are augmented with contiguous areas

satisfying $T_b$<241 K. Cloud systems with 225 K<$T_b$<241 K are also labeled as CCSs if they form a contiguous area of at least
km$^2$. Then the CCSs identified in two consecutive hours are linked if their spatial overlaps are greater than 50%. The
resultant CCSs, precipitation features, and convective core features are used to categorize precipitating systems into MCS,
IDC, and non-convective systems (Feng et al., 2023; Li et al., 2021a). This dataset provides 3D information on the lifecycle
of each MCS/IDC event, including its spatial coverage, radar echo-top heights, precipitation characteristics, convective core

area, and propagation speed. This study further separates tropical cyclones (TCs) and associated precipitation from the
convection dataset using the historical and most recent TCs obtained from the IBTrACS (International Best Track Archive for
Climate Stewardship) version 4.0 dataset (Knapp et al., 2010).

## 2.2 NEXRAD data at KHGX

The 4-km and hourly convection dataset can potentially overlook many isolated convections having lifetime and spatial

extent smaller than these scales. We track and characterize fine-scale convection using radar located in Houston (KHGX). The
original Level II reflectivity is obtained from Amazon Web Services (https://registry.opendata.aws/noaa-nexrad, last access:
Oct 2022) from 2004 to 2017 at approximately 5 minute intervals, and interpolated from antenna coordinates to Cartesian
coordinates at 500 m × 500 m × 500 m ($x \times y \times z$) grid spacing using Py-ART (https://arm-doe.github.io/pyart). Then, we apply
the open-source Python Flexible Object Tracker (PyFLEXTRKR, available on: https://github.com/FlexTRKR/PyFLEXTRKR)

algorithm (Feng et al., 2023) to identify and track convective cells from the gridded radar data.

Convective cells are identified in PyFLEXTRKR using a modified Steiner et al. (1995) algorithm based on horizontal radar
reflectivity texture. We used composite (column maximum) radar reflectivity to compute horizontal peakedness (i.e., the
difference between a grid point reflectivity and its surrounding background reflectivity) and identify convective cells, similar
to the approach described by Feng et al. (2022). Radar reflectivities less than 500 m above terrain are removed to reduce clutter

contamination. The various thresholds used to define convective cells for tracking are the same as those used in Feng et al
(2022). A convective grid point is defined as its difference in composite reflectivity ($\Delta Z$) from the background reflectivity
($Z_{bkg}$, horizontal mean reflectivity within 11 km radius), which is described below:

$$\Delta Z = \begin{cases} 10 \cdot cos\left(\dfrac{\pi \cdot Z_{bkg}}{2 \cdot 60}\right), & Z_{bkg} \geq 0 \; dBZ \\ 0 & Z_{bkg} > 60 \; dBZ \end{cases}$$

The main purpose of the adjusted thresholds is to better identify individual deep convective cell initiation in a variety of

situations (e.g., isolated convection in summer, convective cells embedded in MCSs) compared to those used in Steiner et al.



(1995). Such adjustments are similar to those used for the study of convective cell growth observed during the CACTI (Cloud, Aerosol, and Complex Terrain Interactions) field campaign in central Argentina (Feng et al. 2022; Varble et al. 2021). The 500 m grid spacing of the Cartesian NEXRAD radar data is the same as the attenuation-corrected C-band radar data used in CACTI. The convective grid points are then expanded outward into surrounding grids using a $Z_{bkg}$-dependent radius step function to define convective cells. Lastly, each convective cell is expanded outward by a 5 km radius from the center of the cell to increase the footprints for the convective cells for tracking. Figure 1 shows an example of cell tracking based on KHGX radar reflectivity.



**Figure 1 Example of cell tracking. Shadings are radar reflectivity. Circles are the areas of CCSs and the dots indicate the center of the CCS. Numbers are the cell IDs.**

## 2.3 ERA5 reanalysis

The European Centre for Medium-Range Weather Forecasting 5th generation Reanalysis (ERA5, Hersbach et al., 2020) is utilized to perform the large-scale meteorological pattern classification. The hourly variables for the period of 2004-2017 at horizontal resolution of 0.25°×0.25° are obtained from https://www.ecmwf.int/en/forecasts/dataset/ecmwf-reanalysis-v5 (last





access, Oct, 2022). The horizontal wind fields ($u$, $v$) and specific humidity ($q$) at 925 hPa are used to classify LSMPs, following Song et al. (2019). Once the LSMPs have been identified, other fields including 2-m temperature, geopotential height, and vertical velocity are used to characterize the overall environmental conditions for the LSMPs.

### 2.4 Self-organizing maps

SOM is an artificial neural network employed for cluster analysis, which projects the high-dimensional data to a visually comprehensible two-dimensional map (Vesanto and Alhoniemi, 2000; Song et al., 2019). Hourly $u$, $v$, and $q$ values are normalized by removing the long-term mean and standard deviation at each hour to give each variable equal weights. In the SOM training phase, the initial nodes for SOM clustering are selected from the leading empirical orthogonal functions of the input vector ($u$, $v$, and $q$). Then input vectors are presented on the map to find the best matching unit (BMU), which is the node

with the smallest Euclidean distance to the input vector. The BMU and its neighboring nodes are adjusted toward the input vector to better represent the data distribution. A neighborhood function is applied to determine the number of neighborhood nodes to be adjusted and the strength of adaption, depending on the order number of the current iteration and the distance between the neighborhood node and the BMU.

     The SOM analysis is performed over the domain 15°–50°N, 120°–70°W using the 4-km convection dataset. In contrast to

Wang et al, (2022), which used all hours to train the SOM model, we only use the initial hour of MCS/IDC tracks that produce precipitation in the southeastern Texas area (28°–32°N, 97°–93°W) to focus only on the large-scale environments that are associated with convection initiation in that region. The environment at the initiation is targeted to minimize the effect of convection feedback to the large-scale environment (Song et al., 2019). The convection initiation hours are then grouped into two sets: one only with IDC initiation and another with MCS initiation, so that the large-scale environments associated with

IDC and MCS initiation are separated. We consider the periods of March-May (MAM), June-August (JJA), September-November (SON), and December-February (DJF) to construct a training data set for each season. Choosing an appropriate number of SOM nodes to prescribe requires balancing the trade-off between distinctiveness and robustness (Liu et al., 2023; Liu et al., 2022; Huang et al., 2022). Song et al., (2019) found that clustering the convection associated weather patterns over the Great Plains using four nodes resulted in distinct large-scale environments while minimizing redundant nodes. They also

discovered that the results were not sensitive to the domain size. In this study, a similar domain as Song et al., (2019) is used to conduct SOM analysis in order to capture the interaction between the Bermuda High, GPLLJ and mid-latitude atmospheric waves. Therefore, we chose four nodes for each season. Lastly, the KHGX radar data are projected onto the four LSMPs by matching the hours in each LSMP in summer to investigate the fine-scale convection.





## 3 Results

### 3.1 Precipitation features and associated large-scale environments


To elucidate the role of large-scale environment on precipitation over southeastern Texas (indicated by the purple box in Fig. 2), the climatological mean and anomalies (deviation from the climatology) in wind and moisture at 925 hPa are shown in Figure 2. The GPLLJ, transporting a substantial amount of moisture from the Gulf of Mexico to the southern Great Plains, is a seasonal feature that varies throughout the year, affected by factors including pressure and temperature gradients, upper-

level wave patterns, and land surface properties (Bonner, 1968; Weaver and Nigam, 2008; Yang et al., 2020). During spring and summer, the gradient between the high pressure over the Rocky Mountains and the low pressure over the Gulf of Mexico combined with the warm surface temperature enhances the GPLLJ (Fig. 2a, b). In spring, the Bermuda High sits in the east Atlantic Ocean, with its west flank superimposed on the GPLLJ, resulting in the largest wind speeds (Bonner, 1968; Higgins et al., 1997). In contrast, during summer, the GPLLJ weakens when the Bermuda high dominates the Great Plains. Despite

being slightly weaker in summer, the GPLLJ still transports ample moisture inland. In fall, the GPLLJ tends to weaken even further as the temperature difference between the Gulf of Mexico and Great Plains decreases (Fig. 2c). In winter, the GPLLJ is typically weakest, but can still affect the weather patterns in southeastern Texas (Fig. 2d) (Weaver and Nigam, 2008). At the upper levels (200 hPa), southeastern Texas is located southeasterly ahead of a large-scale trough during spring, fall and winter that, coupled with baroclinic waves, are favorable to upward motion (Fig. 3a, b, d). In contrast, the region is occupied

by a high-pressure ridge in summer which contributes to convection inhibition (Fig. 3b).



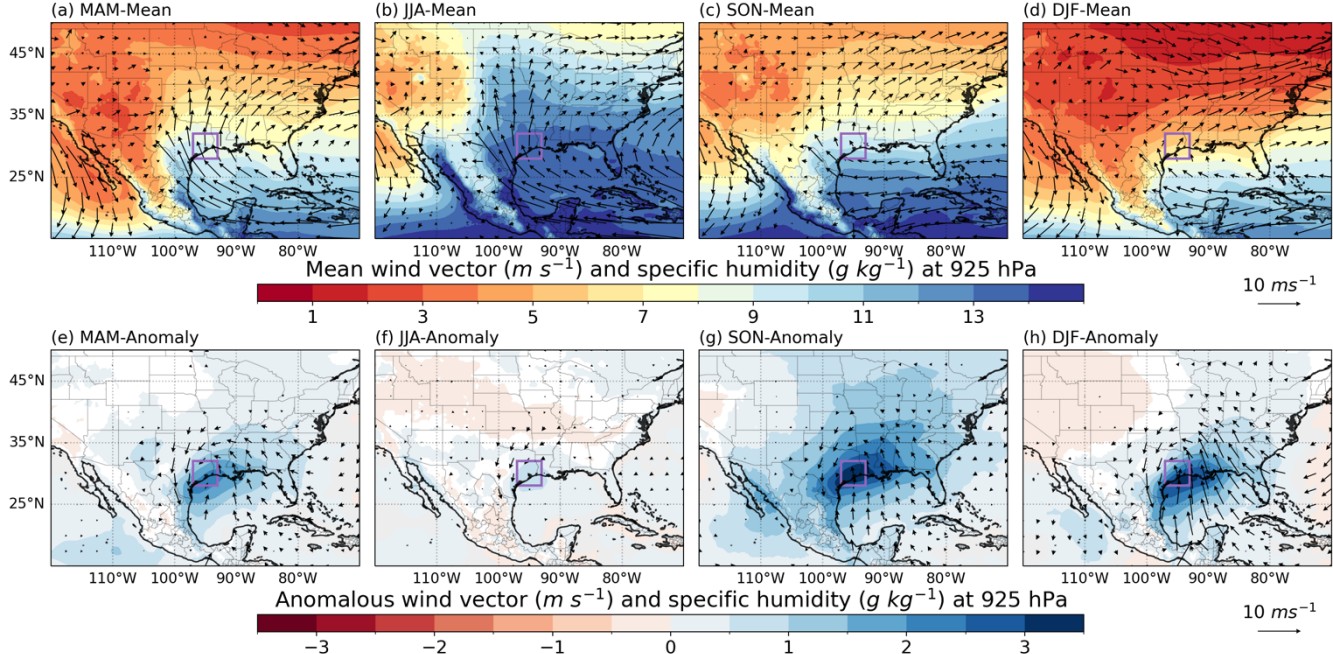

**Figure 2: Mean (a-d) and anomalous (e-h) wind vector (m s⁻¹) and specific humidity (g kg⁻¹) at 925 hPa for each season. The anomaly**
**is the difference between the pattern associated with precipitation in southeastern Texas and the climatology. The southeastern**
**Texas area is marked by the purple box (28°N – 32°N, 97°W – 93°W).**

When precipitation occurs, anomalous cyclonic flow roughly centered over southeastern Texas are observed. The
magnitude of moisture anomaly varies and is found to be weakest in summer and strongest in fall and winter (Fig. 2), consistent
with previous studies (Feng et al., 2016; Geerts et al., 2016; Haberlie and Ashley, 2018). Furthermore, southeastern Texas is
situated between an anomalous upper-level trough and ridge, suggestive of mid-level upward motion. The upper-level
anomalies are strongest in fall and winter, followed by spring, and insignificant in summer (Fig. 3). Local thermodynamic
factors become important in driving summer time convection because of the absence of large-scale anomalies (Wang et al.,
2022).



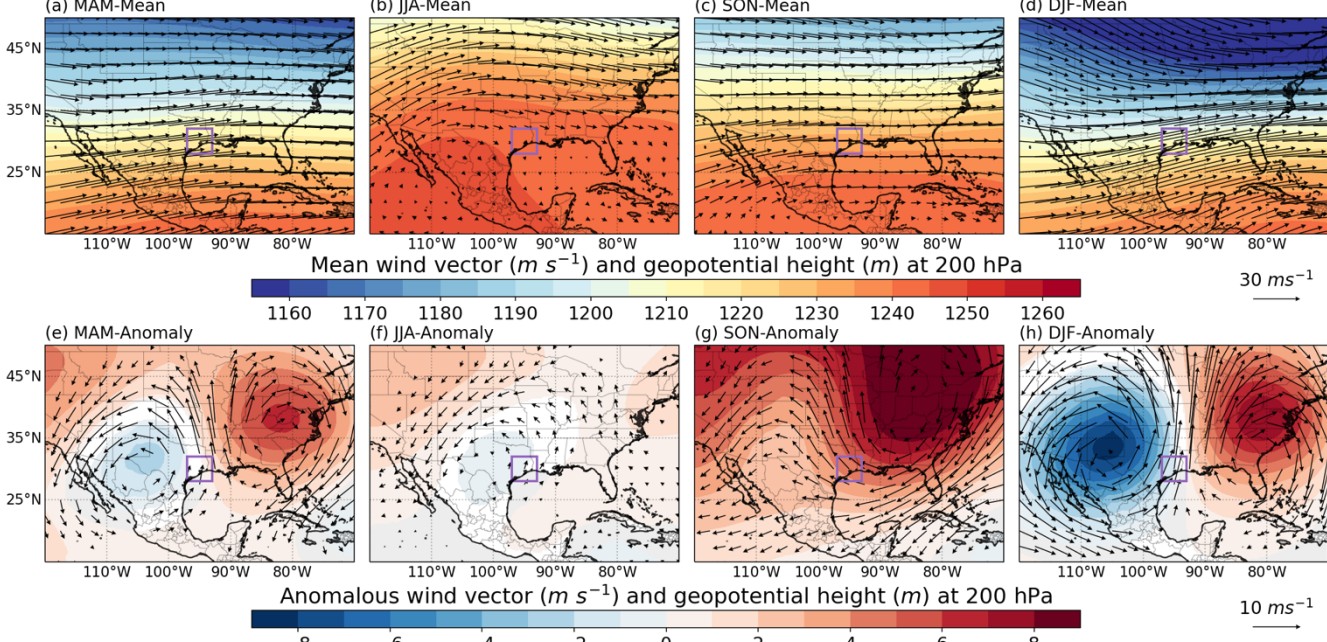

**Figure 3: The same as Fig. 2, except for wind vector (m s⁻¹) and geopotential height (m) at 200 hPa.**

We analyze the statistical precipitation features during four seasons (Fig. 4 and Table 1). The largest precipitation rate (over the entire period) is observed in summer (3.9 mm d⁻¹), followed by fall (3.4 mm d⁻¹), spring (3.3 mm d⁻¹), and winter (2.8 mm d⁻¹). MCSs are the primary contributor to the mean precipitation in all seasons, accounting for 66.5%, 33.4%, 41.1%, and 47.3% of precipitation amount, in spring, summer, fall, and winter, respectively. In general, the relative amount of MCS associated precipitation over southeastern Texas in each season is consistent with the findings over the southern Great Plains in a previous study focusing on long-lived MCSs east of the Rocky Mountains (Feng et al., 2019). In addition to MCS, IDCs contribute to 31.7% summer precipitation and non-convection (NC) contribute to 23.9%, 33.1%, and 37.9% of precipitation in summer, fall, and winter, respectively, especially in the eastern part of the coastal region. TCs contribute to around 10% of summer and fall total precipitation and are rarely observed in spring and winter.





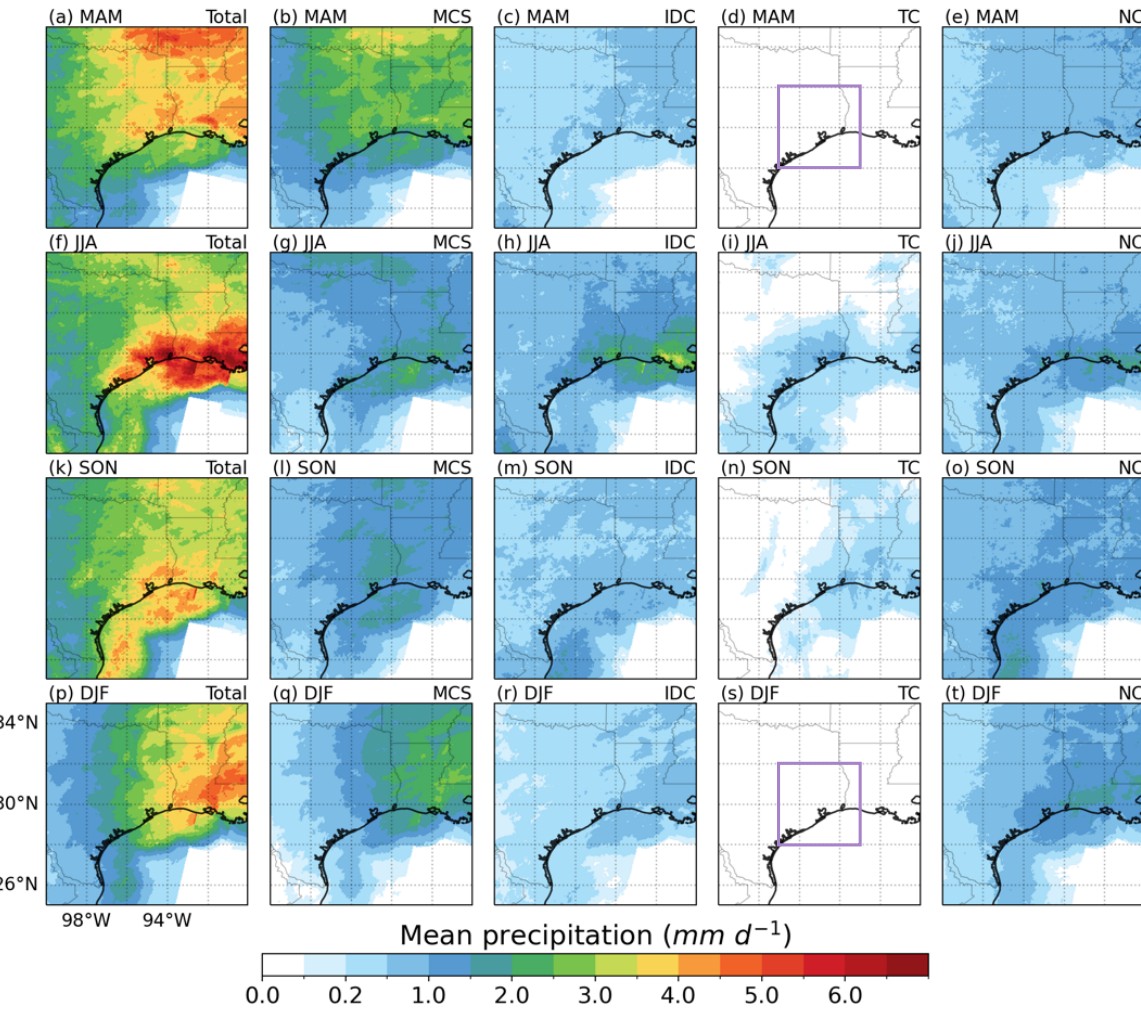

**Figure 4: Mean precipitation amount (mm d⁻¹) in each season and for each precipitation type. The purple rectangle indicates the southeastern Texas area to calculate the statistics. NC stands for non-convection.**

The coastal region experiences more intense precipitation than inland areas. Precipitation intensities are similar in spring, fall, and summer, ranging from 2.8 to 3.0 mm h⁻¹, but weaker in winter (2 mm h⁻¹) (Fig. 5 and Table 1). Of the four precipitation types, MCSs bring intense precipitation in all seasons, ranging from 4.0 mm h⁻¹ to 4.9 mm h⁻¹. IDCs produce intense precipitation in both summer and fall (mean precipitation intensities of 4.0 mm h⁻¹ and 3.9 mm h⁻¹, respectively). NC, with smallest precipitation intensity among the four precipitation types, have a larger contribution to total precipitation in winter than other seasons, resulting in the weakest mean precipitation during winter. TCs generate the most intense precipitation in summer and fall, with an average precipitation intensity exceeding 4.7 mm h⁻¹, but make a small contribution to the total precipitation amount.





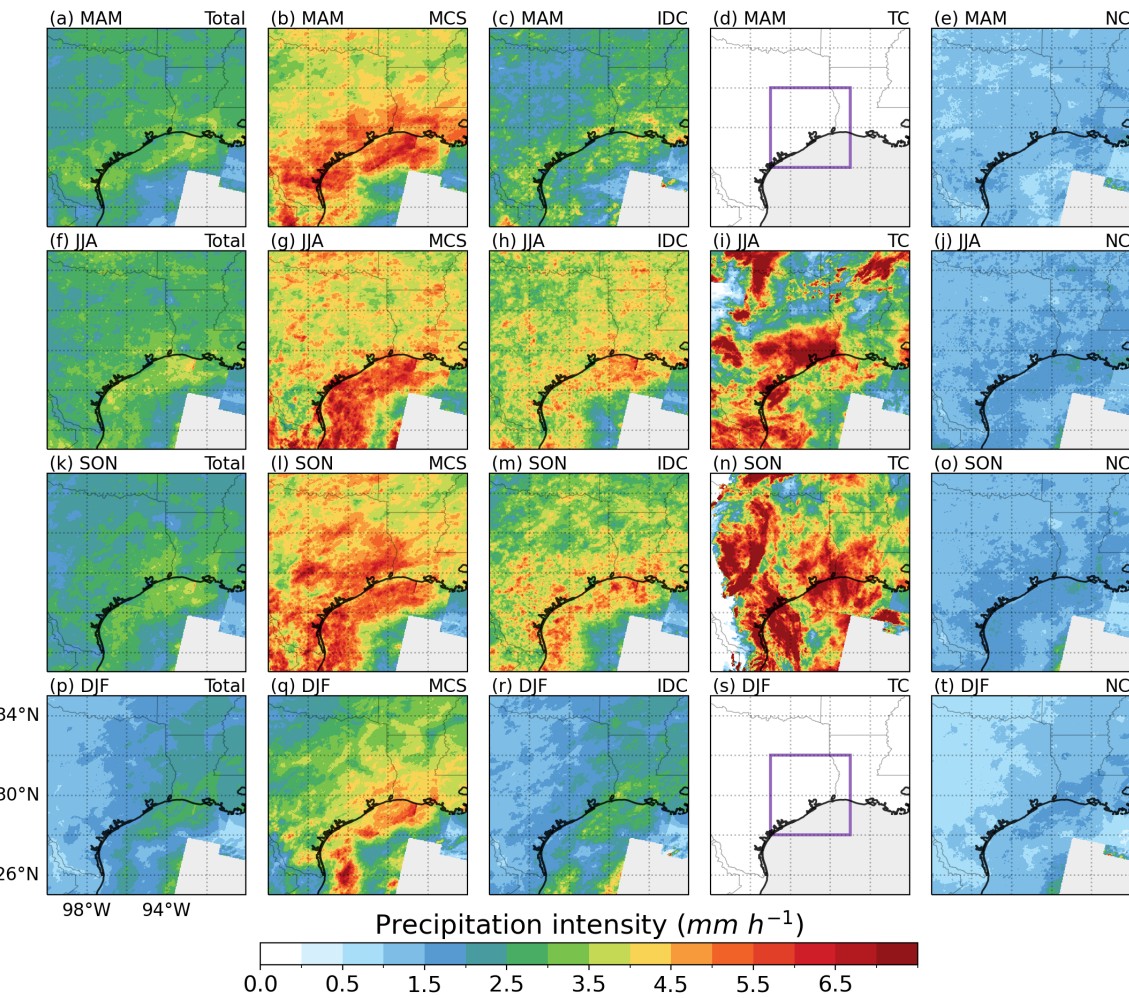

**Figure 5: The same as Fig. 4, but for precipitation intensity (mm h⁻¹).**

To identify seasonal extreme precipitation events, we adopt a definition based on daily/hourly precipitation greater than

99th and 95th percentiles of all days/hours with rainfall using all grid points within the southeastern Texas region, representing the top 1% and top 5%, respectively (Li et al., 2021b). We find consistent statistics using the top 1% and top 5% definitions (Table 1). MCSs contribute the majority of extreme precipitation, particularly in spring accounting for 77.7% to 88.3% of both daily and hourly extremes. In comparison, IDC have a more significant impact on hourly extremes than daily extremes, whereas NCs have greater contributions to the daily metric. The differences in the lifetime of the three precipitation types could explain

the differences found between the daily and hourly metrics (Table 2). MCS commonly have a lifetime of 20 hours with heavy precipitation, causing extremes on both daily and hourly timescales. In contrast, IDC produce intense precipitation for an average of 2 hours leading to a moderate contribution to the daily average. NCs produce light to moderate precipitation for a





few hours to days, therefore, have a significant contribution to the daily metrics. Moreover, TCs in fall and summer can produce intense precipitation from hours to a day, contributing to 7.0% to 36.9% of the daily and hourly extremes, respectively.


**Table 1: Statistics of precipitation features including mean precipitation, precipitation intensity, and the contribution of different precipitation types.**

| Seasons | Precip.* Type | Mean precip. (mm d⁻¹) | Mean precip. intensity (mm h⁻¹) | Contribution to total mean precip. (%) | Contribution to total extreme precipitation (%) | | | |
|---|---|---|---|---|---|---|---|---|
| | | | | | Extreme daily precip. | | Extreme hourly precip. | |
| | | | | | Top 1% | Top 5% | Top 1% | Top 5% |
| Spring | MCS | 2.2 | 4.8 | 66.5 | 77.7 | 75.8 | 88.3 | 83.1 |
| | IDC | 0.5 | 2.7 | 13.7 | 2.5 | 5.4 | 8.6 | 10.2 |
| | TC | 0 | 0 | 0 | 0 | 0 | 0 | 0 |
| | NC | 0.7 | 1.2 | 19.8 | 19.8 | 18.8 | 3.1 | 6.7 |
| Summer | MCS | 1.3 | 4.4 | 33.4 | 34.5 | 39.8 | 45.4 | 41.3 |
| | IDC | 1.2 | 4.0 | 31.7 | 6.2 | 15.5 | 35.7 | 37.1 |
| | TC | 0.4 | 4.7 | 11.0 | 36.9 | 22.2 | 14.0 | 12.4 |
| | NC | 0.9 | 1.5 | 23.9 | 22.4 | 22.5 | 4.9 | 9.2 |
| Fall | MCS | 1.4 | 4.9 | 41.1 | 41.9 | 42.7 | 54.5 | 49.3 |
| | IDC | 0.6 | 3.9 | 18.9 | 6.5 | 11.0 | 26.6 | 26.2 |
| | TC | 0.2 | 5.0 | 6.9 | 12.3 | 10.1 | 7.1 | 7.0 |
| | NC | 1.1 | 1.6 | 33.1 | 39.3 | 36.2 | 11.8 | 17.5 |
| Winter | MCS | 1.3 | 4.0 | 47.3 | 49.9 | 49.6 | 70.0 | 64.1 |
| | IDC | 0.4 | 2.1 | 14.8 | 4.4 | 8.9 | 14.6 | 16.6 |
| | TC | 0 | 0 | 0 | 0 | 0 | 0 | 0 |
| | NC | 1.1 | 1.2 | 37.9 | 45.7 | 41.5 | 15.4 | 19.3 |

*precip. stands for precipitation. MCS, IDC, TC, NC stand for mesoscale convection system, isolated deep convection, tropical cyclone, and non-convection, respectively. The top 1% and 5% daily/hourly precipitation are calculated as precipitation greater than 99ᵗʰ and 95ᵗʰ on the respective timescale.


### 3.2 Large-scale environments associated with MCS and IDC initiation in different seasons

In this section, we dive into the connection between the large-scale environment and MCS and IDC events. Overall, MCSs more frequently occur in spring and summer than fall and winter, whereas IDC occurrences are concentrated in summer and early fall (Table 2). LSMPs associated with MCS initiation ensembles three frontal systems patterns and an anticyclone pattern in all seasons. These LSMPs are similar in spring, fall and winter, but they are much weaker in summer. The similar four LSMPs are found to be associated with IDC initiation, but different patterns are found in summer. Therefore, we focus on comparisons of LSMPs in spring and summer and between MCS and IDC.

In spring, three LSMPs associated with frontal systems, namely pre-front, front and post-front LSMPs, account for 27%, 23%, and 22% of MCS occurrences, respectively, and 30%, 20%, and 27% of IDC occurrences (Fig. 6a – 6c), respectively. During the pre-front LSMP, anomalous southerly winds dominate the southern Great Plains and extend inland, while weak westerly winds prevail east of the Rocky Mountains (Fig. 6a). This resembles a synoptic front, or trough when the front is absent, between the Rocky Mountains and the southeastern Texas, accompanied by a strong moisture gradient across the front



(Fig. 6a and Fig. 7a). The strong moisture gradient is also known as the dryline which often favors convective storm initiation (e.g., Hoch and Markowski, 2005). During the front LSMP, wet anomalies shift southeast with slightly clockwise tilting

coinciding with anomalous northwesterly winds and low moisture west of southeastern Texas (Fig. 6b and Fig. 7b). As a result, an intensified front resides over southeastern Texas, with converging moisture and enhanced mid-level lifting (Fig. 8b). During the post-front LSMP, the anomalous frontal system is further east, positioning the front to the southeast edge of southeastern Texas (Fig. 6c and Fig. 7c). The three frontal LSMPs depict different stages of a frontal passages and locations of dryline, the baroclinic forcing near the front lifts moist parcels and favors convection initiation.

The fourth LSMP, the anticyclone, does not appear in the anomalous patterns that were selected to include all convective events (Fig. 2e), although it accounts for 28% and 23% of all MCS and IDC occurrences, respectively. The anticyclone LSMP exhibits distinct circulations from frontal-related LSMPs, with anomalous high pressure dominating southeastern Texas (not shown). The anomalous northeasterly winds transport moist air over the east slope of the Rocky Mountains, which could trigger convection through orographic lifting (Figs. 6d and 7d).


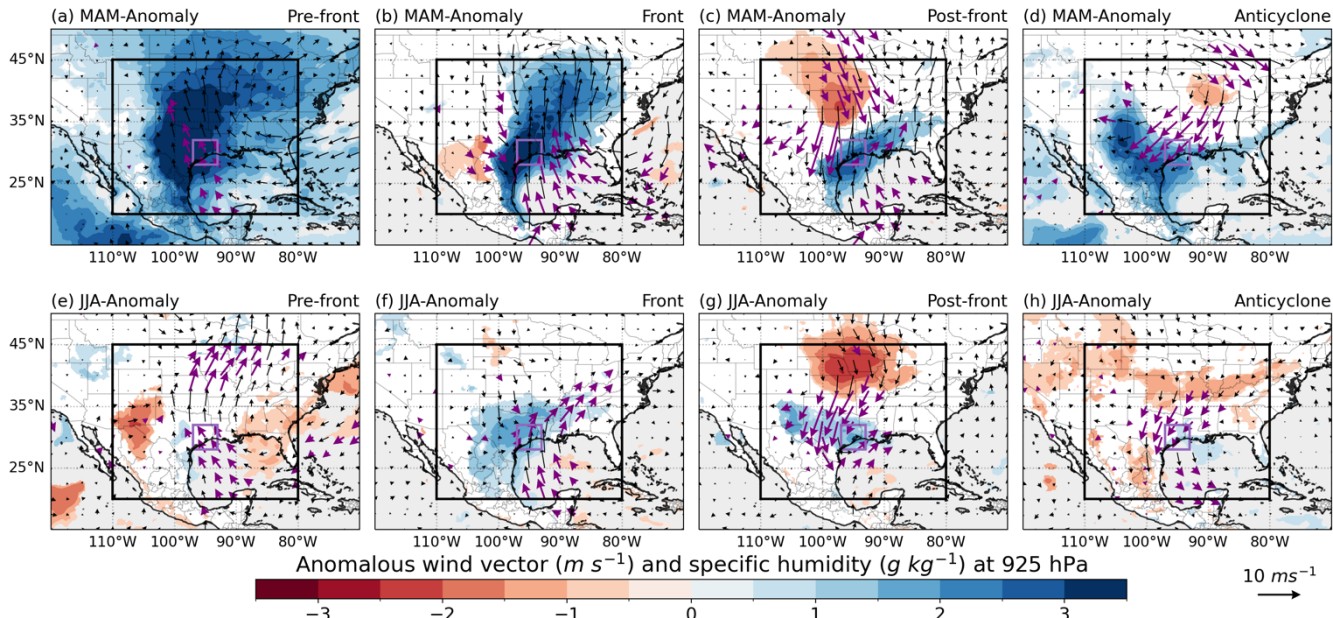

**Figure 6: Anomalous wind vector (m s⁻¹) and specific humidity (g kg⁻¹) at 925 hPa for spring (a-d) and summer (e-h) associated with MCS initiation. Significant vectors at the 5% level according to the student's t-test are shown in purple. Only significant specific humidity anomarlies are shown. The black box indicates the region for SOM analysis.**

During the summer months, the four LSMPs occur with almost equal frequency, display similar patterns as those observed during spring, but smaller in magnitude (Fig. 6e-h). The frontal lifting mechanism can easily trigger MCS initiation in a moist and warm summer environment. The anticyclone LSMP is also present in the summer, but without significant moisture anomalies and air stacking which are seen in spring. Regarding IDC initiation, two groups of LSMP are identified: two front-





related and two anticyclone-related patterns. The pre-front and post-front LSMPs account for 34% and 15% of the IDCs,
respectively (Fig. 7e, g). The remaining 34% and 17% of IDC initiations are associated with the anticyclone or weak-anticyclone conditions, respectively.

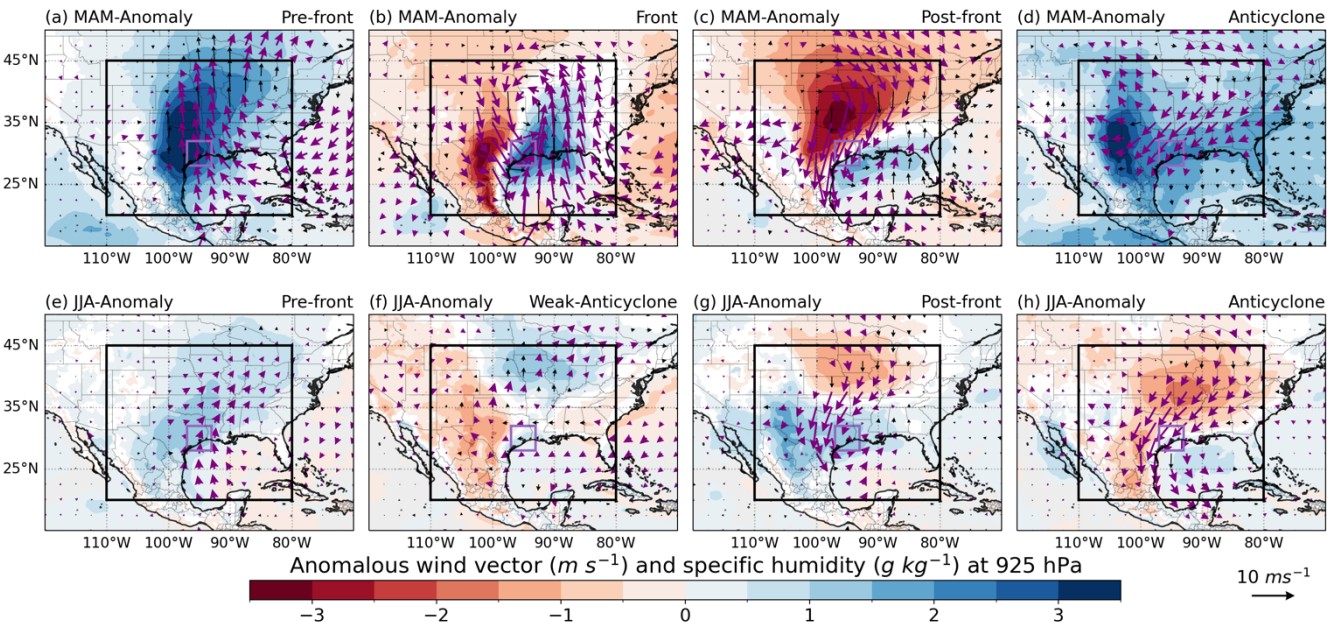

**Figure 7: The same as Fig. 6, but for IDCs.**

The frontal system and orography forcing mainly provide the lifting force in spring, which can be observed from the middle atmosphere large-scale upward motion. The anomalies of 500-hPa vertical velocity are greater than -0.2 Pa s$^{-1}$, suggesting a strong upward motion (Fig. 8). The upward motion area also shifts eastward and rotates clockwise from pre-front to post-front LSMP, whereas the upward motion area is confined over the eastern slope of the Rocky Mountains to the west of southeastern Texas with anticyclone conditions. In contrast, significant mid-tropospheric upward motion is only found during frontal-related
LSMPs in summer. No large-scale lifting mechanism is observed during anticyclone LSMPs, where mesoscale and local factors such as sea-breeze forcing may trigger convection.





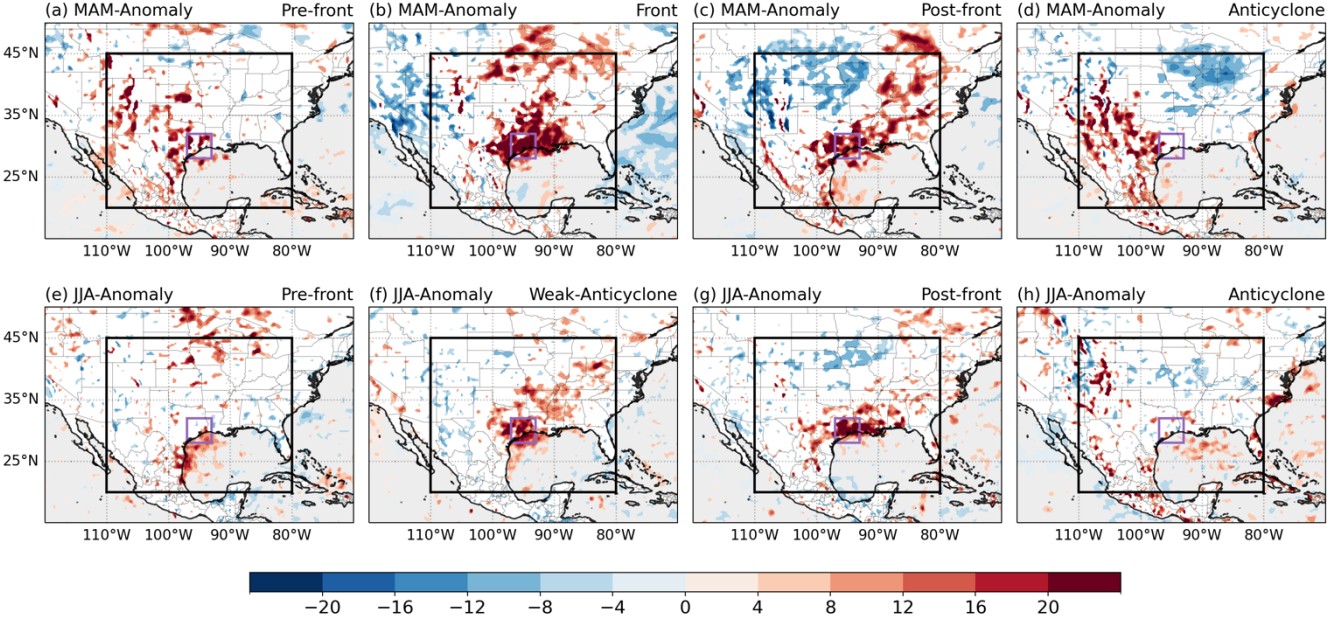

**Figure 8: The same as Fig. 6, but for 500-hPa vertical velocity (× -100 Pa s$^{-1}$). Only significant anomalies at the 5% level according to the student's t-test are shown.**

The composite analysis of the four LSMPs in fall and winter are generally similar to those in spring, except for less significant anomalies for winter MCS. It is worth noting that the mid-tropospheric ascent anomalies are strongest in winter, which is possibly needed to support deep convection in an environment with the lowest moisture and instability.

### 3.3 Tracking the convective systems

In this section, we examine the characteristics of convection affecting southeastern Texas, including initiation location, travel path, lifetime, precipitation duration, cold cloud system area, and rain rate. This analysis includes 180, 175, 103, and 80 MCSs and 2917, 9460, 4613, and 2249 IDC in spring, summer, fall, and winter, respectively. All those track features are included in the convection dataset developed by Li et al. (2021a). In spring, a significant proportion of MCSs, ranging from 59% to 82% depending on the LSMP, initiate outside southeastern Texas and propagate into the region, whereas the remaining systems initiate locally (Fig. 9a-d). With spring frontal-related LSMPs, the preferred location for MCS initiation shifts from central and western Texas to the southeastern coast as the frontal system progresses. We observe a shift in the MCS travel direction from southeastward in pre-frontal conditions to northeastward/eastward in frontal/post-frontal conditions, consistent with the clockwise tilting of the steering-level winds associated with the frontal systems. Consequently, MCSs tend to decay in an area close to southeastern Texas in the west-east direction during the pre-frontal LSMP, while decaying farther eastward during the frontal and post-frontal LSMPs. Over 90% of the spring MCSs travel eastward, even in anticyclone conditions, aligning with the mean background wind direction. In summer, MCSs initiate in a narrow longitude band (100°W–90°W) on both the west and east sides of southeastern Texas. Depending on season, 50%–85% of MCSs initiate outside and propagate




westwards or eastwards to southeastern Texas. Over 70% of the front-related MCSs and 55% of anticyclone-related MCSs travel eastward. In fall and winter, MCSs initiate close to southeastern Texas and travel further east, especially during the front-related conditions, resembling the characteristics of both spring and summer except with lower occurrence.




**Figure 9: Track path of MCSs. Black and red dots indicate the location of initiation and decay, respectively. The purple box indicates the southeastern Texas region. MAM, JJA, SON, and DJF represent the boreal spring, summer, fall, and winter, respectively.**

The travel time for MCSs to reach southeastern Texas varies across seasons, with approximately 9 hours in spring, 8 hours in summer and fall, and 4 hours in winter (Table 2). Apart from the initiation location, the travel time can also be influenced by the faster background wind speeds during cold seasons than warm seasons. The difference in movement speed also contributes to the longest duration of precipitation over southeastern Texas in summer, while the shortest duration is in winter. Comparing LSMPs within each season, MCSs associated with pre-frontal and front LSMPs move more quickly through southeastern Texas than post-frontal and anticyclone LSMPs. This behavior can be attributed to the fact that the former LSMPs





are associated with southerly anomalies that are superposed on prevailing southerly winds, whereas the latter LSMPs involve northerly anomalies that decrease the speed of storm advection.

Strong seasonal variations are observed in the size of MCS CCS and precipitation feature (PF). Summer and fall have the smallest CCS, which are nearly twice as large, on average. in the spring and winter. The MCS PF, predominantly stratiform area, is smallest in summer, comparably larger in spring and fall, and largest in winter (Table 2). In summer, local factors play

a more important role in MCS initiation and, consequently, creating relatively larger convective area (7400–107000 km$^2$ across LSMPs) and smaller stratiform area (243000–309000 km$^2$), and smaller differences in CCS area across the LSMPs. Conversely, in winter, the large-scale environment becomes the predominant driver to trigger MCS, resulting in relatively small convective area (ranging in 51000–84000 km$^2$ across LSMPs) and large stratiform area (489000–863000 km$^2$). With winter anticyclone condition, the stratiform area is approximately twice as large as other LSMPs. In spring and fall, both convective and stratiform

precipitation area are generally lager during pre-front and front LSMPs than post-front and anticyclone LSMPs.

**Table 2. Statistics of the track features of MCS and IDC in four seasons**

|  | MCS | | | | | IDC | | | | |
|---|---|---|---|---|---|---|---|---|---|---|
|  | MAM | JJA | SON | DJF | Ann | MAM | JJA | SON | DJF | Ann |
| Number of events | 180 | 175 | 103 | 80 | 538 | 2917 | 9460 | 4613 | 2249 | 19239 |
| Lifetime (h) | 26.6 | 23.1 | 28.3 | 28.15 | 26.0 | 2.2 | 2.3 | 2.1 | 2.3 | 2.2 |
| Time for CCS centroid reaching southeastern Texas (h) | 9.1 | 8.1 | 8.2 | 4.4 | 7.9 | 0.4 | 0.2 | 0.3 | 0.5 | 0.3 |
| Duration of precipitation over the southeastern Texas (h) | 5.6 | 7.5 | 6.0 | 4.0 | 6.1 | 1.4 | 1.6 | 1.4 | 1.2 | 1.5 |
| CCS area ($10^3$ km$^2$) | 211.1 | 114.3 | 153.8 | 238.2 | 172.7 | 10.3 | 4.3 | 5.2 | 1.6 | 6.8 |
| PF convective area ($10^3$ km$^2$) | 10.2 | 9.1 | 10.5 | 6.9 | 9.4 | 0.6 | 0.5 | 0.6 | 0.7 | 0.6 |
| PF stratiform area ($10^3$ km$^2$) | 43.0 | 27.6 | 46.7 | 57.9 | 40.9 | 1.8 | 0.8 | 1.2 | 2.9 | 1.3 |
| PF convective rain rate (mm h$^{-1}$) | 7.8 | 8.3 | 8.7 | 6.1 | 7.9 | 6.7 | 9.1 | 9.2 | 5.8 | 8.4 |

Over 94% of IDCs initiate and end locally within southeastern Texas (Fig. 10). Although red and black dots are paired, the red dots are plotted last which makes the box looks more red. Across all seasons, the average travel time to southeastern Texas

ranges from 0.21 to 0.45 hours and the duration of precipitation over southeastern Texas spans from 1.23 to 1.57 hours. IDCs have much a smaller spatial extent than MCS, in terms of convective and stratiform precipitation area, but produce comparable mean convective rain rate compared to MCS (Table 2). Across LSMPs, post-front associated IDCs have the largest CCS area in spring, fall and winter and stratiform cloud area in all seasons.





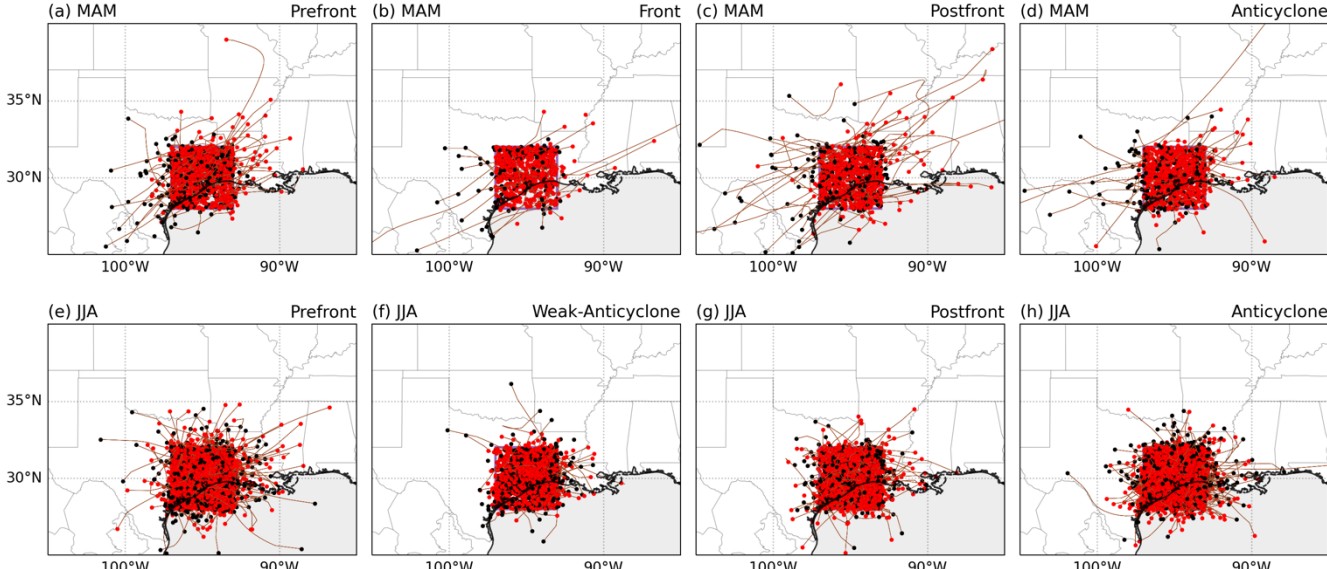


**Figure 10: The same as Figure 9, except for IDC.**

### 3.4 Houston Metropolitan Area

Many of the isolated convective systems have short lifetime and small spatial extent, which scale is not fully resolved by the 4-km and hourly convection dataset. The ~5 minutes and 500 m resolution convective cell tracking data at the KHGX site

is used to study the fine-scale features of IDC near Houston. We project the radar-based tracking data onto the four LSMPs by matching the hours belonging to each LSMP. Almost all MCSs initiate outside the small area covered by radar. Therefore, the hours with MCS initiation are excluded to avoid cells that advected into this area. Among different seasons, summer stands out with the highest frequency of IDC, primarily driven by local factors. Therefore, we select summer as an example to explore the initiation location and time of IDCs near Houston. The percentage of IDCs initiation associated with each LSMP slightly

differs from those based on the hourly dataset, as the KHGX data includes convection that persists for less than one hour and within a smaller area. We find that 40% of IDCs occurred during pre-frontal, 15% during weak-anticyclone, 21% during post-frontal, and 24% during anticyclone LSMPs.

During pre-frontal and post-frontal LSMPs, the combination of warm surface temperatures, abundant low-level moisture, and large-scale lifting creates favorable conditions for triggering of convection both offshore and onshore. (Fig. 11a, c). The

anomalous southerly winds in pre-front LSMP bring the moist flow further inland, whereas the northerly anomalies during post-front LSMP restrict convection to the coastline. In contrast, when an anticyclone dominates the area, convection is primarily triggered by the sea-breeze circulation, where the onshore branch of the circulation initiates convection. Consequently, we find higher occurrences of IDCs initiating onshore rather than offshore (Fig. 11b, d).



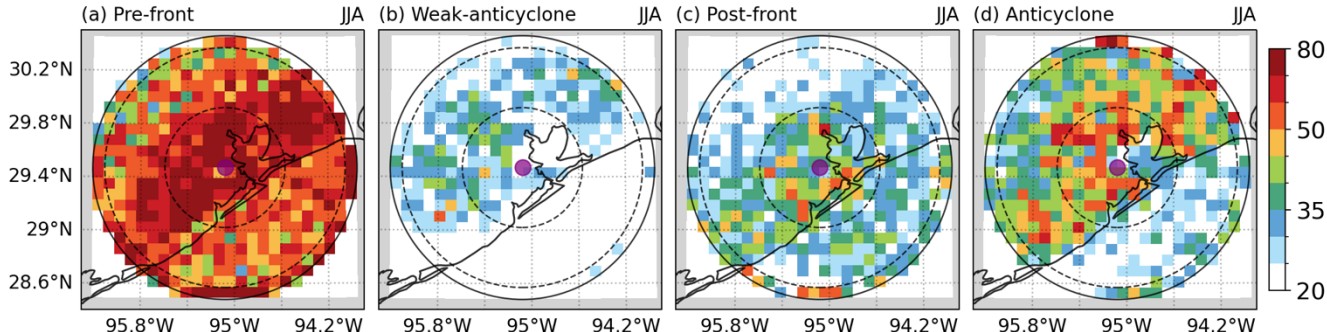


**Figure 11. Number of IDC initialed during each LSMPs in summer. The purple dot indicates the location of the KHGX site. The solid and dashed circles indicate radii of 110 km, 100 km, and 50 km.**

The frontal systems move eastward along with the baroclinic wave passing over southeastern Texas, which can occur at any time of day (Feng et al., 2019). However, the local thermodynamic factors show a strong diurnal cycle. Consequently,

IDCs more frequently initiate between 10:00 AM to 3:00 PM local time (Fig. 12), which is likely due to increased surface heating during the early afternoon. When a baroclinic frontal zone dominates the southeastern Texas even small amounts of convective available potential energy (CAPE) can trigger convection, resulting in a peak IDC initiation near noon. In contrast, when an anticyclone dominates southeastern Texas, IDC initiation requires higher CAPE to overcome convective inhibition (CIN), leading to a peak occurrence in the afternoon. As a result, IDCs associated with more favorable large-scale environments

occur approximately 3 hours earlier than those associated with anticyclones.

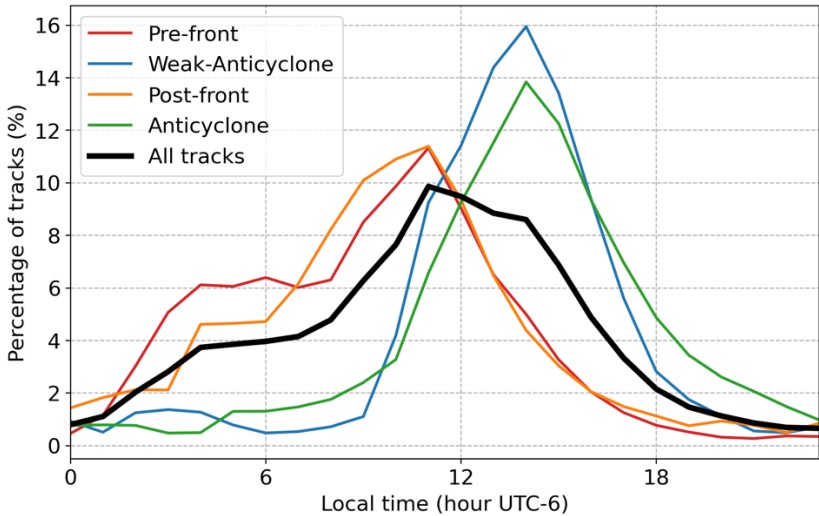

**Figure 12. Percentage of convection initiation at each hour of a day. The percentage is related to the number of tracks for each LSMP.**





The diurnal variation of IDC initiation in spring, fall, and winter are in smaller amplitude than summer (not shown).
Relatively more IDC are trigged in the nocturnal and early morning especially in spring and winter. The large-scale
environment has more important role in triggering IDCs by providing favorable lifting and low-level moisture convergence
that can trigger convection outside local afternoon hours. The diurnally varying thermodynamic forcing slightly increases
convection occurrence in the early afternoon.

## 4 Conclusion

In this study, the characteristics of convection that produce precipitation over the southeastern Texas are analyzed using
13-years of high-resolution observations and reanalysis. The results reveal that MCSs make significant contributions to both
mean and extreme precipitation, with IDCs and TCs also playing a role in generating intense precipitation during the summer
and fall seasons. The contribution of MCS to seasonal precipitation over the southeastern Texas area is consistent with statistics
in previous studies focusing on the southern Great Plains (Feng et al., 2019; Song et al., 2019; Li et al., 2021a). Our findings
also highlight that IDC contribute to one-third of the summer precipitation and over 35% of the hourly extreme precipitation
over the coastal Houston metropolitan area.

The analysis linking convection initiation to weather pattern reveals that both MCS and IDC initiation are predominantly
influenced by large-scale lifting and low-level moisture convergence during spring, fall, and winter. In contrast, summer
convection can even be triggered under weak large-scale circulation anomalies, since the strong surface instability and
abundant moisture provide favorable convective conditions.

Through a breakdown of the convection-associated large-scale environment using SOM analysis, we find the initiation of
convection are associated frontal-related weather patterns and an anticyclone pattern. The frontal-related patterns are more
profound in spring, fall, and winter than summer, which are further differentiated by the stage of frontal passages. The
circulation patterns of the frontal-related LSMPs are consistent with those based on all forms of convection (Feng et al., 2019).
These patterns are characterized by baroclinic waves and low-level moisture convergence, which serve as primary triggers for
convection. Additionally, we find that deep convection can occur even under unfavorable large-scale conditions. An
anticyclone LSMP features northeasterly anomalies that push moist air towards the eastern slope of the Rocky Mountains,
creating an orographic lifting mechanism. In summer, the SOM analysis reveals two groups of weather patterns associated
with either favorable or unfavorable circulations for convection initiation. The opposite circulation anomalies in these groups
offset each other, resulting in a weak overall anomaly when considering all convections.

The large-scale environment impacts the distinct spatial distribution of MCS initiation. In spring, MCSs affecting
southeastern Texas frequently originate between the Rocky Mountain and southeastern Texas and travel eastwards, while
summer MCS initiation concentrates close to Houston. During front-related conditions, MCS initiation location shifts from
central and western Texas to southeastern Texas. Even in anticyclone conditions, MCSs initiate in the west and decay after
moving eastward, aligning with the mean background wind direction. Fall and winter exhibit reduced MCS occurrence but



share similar path features of spring and summer. On average, MCSs initiated remotely require approximately 8 hours of travel time before reaching southeastern Texas, while IDCs are predominantly initiated locally and have a significantly shorter travel time of around 15 minutes.

By analyzing the NEXRAD radar data near Houston, we find that IDC show a higher frequency of occurrence along the Texas coast, with the most occurrence during pre-frontal condition. Furthermore, IDCs exhibit a distinct diurnal pattern, with a peak frequency during the early afternoon. IDCs associated with more favorable frontal-related LSMPs occur approximately 3 hours earlier in the diurnal cycle, peaking at local noon, than those associated with anticyclones.

This study provides insights into the characteristics and spatiotemporal patterns of convective systems under different large-scale environments over the southeastern Texas. Understanding the role of large-scale environment helps tease out the impact of local factors. Our results also offer guidance for selecting cases when the local factors play a dominate role in the initiation of different types of convection.

## Competing interests

At least one of the (co-)authors is a member of the editorial board of Atmospheric Chemistry and Physics.

## Acknowledgement

This study was supported by the U.S. Department of Energy Office of Science Biological and Environmental Research as part of the Atmospheric Systems Research (ASR) Program. The Pacific Northwest National Laboratory is operated by Battelle for the U.S. Department of Energy under Contract DE-AC05-76RLO1830. Computing resources for the simulations were provided by the Environmental Molecular Science Laboratory's (EMSL) on its computational cluster Deception. We would like to thank Jerome D Fast and Colleen M Kaul for review earlier versions of this manuscript.

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
