# Peer review of "Tracking precipitation features and associated large-scale environments over southeastern Texas"

_EGUsphere, 2024_

## Referee Comment (RC2)

Review of "Tracking precipitation features and associated large-scale environments over southeastern Texas" by Y. Liu et al.

*General Comments:*
This paper examines the characteristics of convection systems, i.e., mesoscale convective systems (MCSs) and isolated deep convection (IDC), and their associated large-scale meteorological patterns (LSMPs) over southeastern Texas in different seasons from 2004 to 2017. The tracks of MCSs and IDC in the region and the diurnal cycle of IDC in the Houston metropolitan area are also explored. While the study holds great potential for enhancing the understanding of the spatiotemporal features of convective systems in southeastern Texas, the paper would be more convincing with improved clarity in methodology and analysis.

*Specific comments:*
1. I think the motivation of the study is not quite clear (lines 42-47). In addition to extending previous studies in summer to other seasons, it would be more interesting to highlight what's new in this study and what specific research questions this paper tries to address. It would also be helpful to briefly review previous findings about MCSs and IDC in this region.

2. The 4-km convection dataset is the key dataset used in this study (section 2.1). It would be helpful and informative to briefly explain how MCSs, IDC, and non-convective systems are identified in the dataset and discuss any uncertainties related to the methodology. Adding information about how to separate tropical cyclone-induced precipitation from total precipitation in the dataset (lines 70-73) would be helpful, too.

3. Similarly, it is preferred to add some discussion about the potential uncertainties associated with the KHGX reflectivity and the method to identify convective cells, if any.

4. The usage of 'the initial hour' (line 120) is not very well explained. Is it exactly one hour (or a few hours) ahead of the initialization of convective systems or just the first hour of the convection? If it is just an hour before or at the time of convection, is this time frame too short or sufficient to capture the large-scale features? Have you examined a few hours before the occurrence of MCSs and IDC? Are the results similar or different?

5. In section 3.2, self-organizing maps (SOM)-identified large-scale meteorological patterns (LSMPs) are categorized into three frontal-system patterns and an anticyclone pattern for MCSs and two frontal-related and two anticyclone-related patterns for IDC, but without explanation about why and how these categories are defined. Taking Fig. 7h as an example, the circulation in the black box (purple arrows) shows a somewhat cyclonic pattern, but it is labeled as a type of 'Anticyclone' pattern.

6. It's not very clear why the GPLLJ 'weakens' during summer (line 144). Does this refer to a specific section of the jet? Previous studies showed that the GPLLJ peaks in June-July (e.g., Weaver and Nigam 2008). The meridional winds over the Great Plains region also appear stronger in JJA than in MAM in Fig. 2a-b.

7. Specific humidity at 925 hPa is used in the analysis in different places of the paper (e.g., Figs. 2, 6, 7). While this is the level close to the core of the GPLLJ and probably well reflects low-level moisture transport, technically, precipitation is more directly related to vertically integrated moisture convergence (and also surface evapotranspiration). I wonder if you have tried vertically integrated moisture flux or moisture convergence in the analysis and if you get similar results.

8. In Section 3.4, I'm curious if there are also a lot of IDC events during the fall as shown in Table 2. It would be nice to have results (like Fig. 11) from other seasons in the supplementary materials.

*Technical corrections:*
1. Fig. 2, it's better to mask out wind vectors and specific humidity over the mountainous region where the surface is above the 925 hPa level. In the caption, it says "precipitation" but the text indicates it's convective precipitation (lines 120-121). Can you please clarify?

2. Figs. 6-7, can you please add the frequency (%) to each LSMP pattern to the figures?

3. Fig. 12, is this for summer or all the seasons?

4. Lines 319-320, which figure do you refer to?

---

## Author Comment (AC1)

**Reviewer #1**

With 13-year high-resolution observations and reanalysis data, this study investigated the characteristics and spatiotemporal patterns of different types of convective systems over southeast Texas, and it found that mesoscale convective systems (MCSs) are essential to both mean and extreme precipitation in all seasons, while isolated deep
5    convection (IDC) is more important for the intense precipitation during summer and fall. And with the help of self-organizing maps (SOMs), it demonstrated frontal-related and anticyclones large-scale meteorological patterns (LSMPs) for convection. Then it discussed the lifecycle of MCSs and IDC over southeastern Texas. Generally, the manuscript is well written, and the results are reasonable. My concerns are listed in the following.

We thank the reviewer for their thoughtful and constructive comments and suggestions, which has substantially
10    improved the quality of the manuscript. We have addressed all the reviewer's concerns and revised the manuscript accordingly. Our point-by-point responses are in blue and the modifications to the manuscript are quoted in green.

1. Line 113-114:How many leading empirical orthogonal functions of the input vector are used for the initial nodes for SOM clustering?

    The initial nodes are the first guess of the four SOM nodes, therefore, four leading EOF modes are used. It has
15       been clarified in the revision.

2. Line 149: What are the baroclinic waves observed in Fig. 3a, c, d?

    We agree with the reviewer that the baroclinic waves cannot directly be observed in Figure 3a, c, d. Therefore, this statement has been removed in the revision.

3. Line 215-216: I cannot see the weak westerly winds prevails east of the Rocky Mountains (Fig. 6a).

20    Thanks for pointing this out. It should be weak anomalous winds east of the Rocky Mountains. We have modified the manuscript.

4. Line 223-224: If the three frontal LSMPs depict different stages of a frontal passages, they should be continuous in time, any evidences to show it? According to the different explanation ratios of 27%, 23% and 22% of MCS occurrences by the three LSMPs, it seems that some of them do not appear successively.

25    The reviewer has highlighted an important consideration. The three frontal LSMPs are expected to reflect the large-scale atmospheric conditions that could be associated with a surface front, which propagate from west to east and exhibit a clockwise tilt. However, this may not always hold true. For example, a front situated to the west of the study area, indicative of a pre-frontal condition, may dissipate before it traverses the region. As a contrast, a front located to the east of the study area, representative of a post-frontal situation, may materialize without having
30    visibly moved in from the west. To avoid confusion, we have revised this sentence as following (Line 256-257):

    *The three frontal LSMPs depict different locations of front and dryline. The baroclinic forcing near the front lifts moist parcels and favors convection initiation.*

5. Line 237-238: As for the statement that there are no significant moisture anomalies and air stacking in summer than in spring, it's a little strange that summer environment should be more moist and warmer as mentioned by the authors.

35

With relatively dry condition in spring, the development of convection requires extra moisture transported from the east where is relatively wet. This indicates an anomalous moisture transport as depicted in Figure 7d. As a contrast, the summer environment is moister, the convection initiation is not strongly linked to anomalous moisture transport (Figure 7h).

40  6. Line 322-323: How to conclude that the convection is primarily triggered by the sea-breeze circulation when an anticyclone dominates the area? Does the convection usually occur in the daytime due to the land-sea thermal contrast? It seems to be true according to Fig. 12. However, on the other hand, since the sea-breeze circulation can appear in all the four seasons, why are more IDC triggered in the nocturnal and early morning especially in spring and winter (Line 340)?

45

Under anticyclone conditions, the initiation of convection is associated with enhanced land-sea thermal contrast (Fig. R1 b and d), suggesting a strong sea-breeze circulation.

It is true that the convection usually occur in the daytime, frequency peaking in the early afternoon when the land-sea thermal contrast reaches its maximum. However, the peaking hour of convective occurence is modulated by both the intensity of the sea-breeze circulation and the large-scale atmospheric conditions. In summertime

50  anticyclone conditions, the sea-breeze circulation emerges as the major trigger mechanism, leading to a peak in the afternoon. In other seasons, large-scale dynamical forcing plays a major role in triggering the convection, while the diurnally varying thermodynamic forcing still exerts a modest influence, slightly increases convection occurrence in the early afternoon.

The main confuse arises from the sentence in the original version of the manuscript: *"Relatively more IDC are*

55  *trigged in the nocturnal and early morning especially in spring and winter"*. We have rewritten it in Line 378-380 as:

*While IDC also frequently initiate in the afternoon during spring and winter, there are more IDC occur in the nocturnal and early morning compared to summer.*

[Figure]

**Figure R1 Summertime 2-m temperature (ºC) for each LSMP associated with IDC initiation.**

60    7.   Line 361-363: is the orographic lifting mechanism unfavorable large-scale condition for the deep convection?

Thanks for pointing this out. We have revised this sentence in Line 400-402 as:

*Additionally, we find that deep convection can occur even under unfavorable large-scale **meteorological** conditions. The northeasterly anomalies associated with the anticyclone push moist air towards the eastern slope of the Rocky Mountains, creating an orographic lifting mechanism.*

65    8.   Line 149: (Fig. 3a, b, d) should be (Fig. 3a, c, d); Line 214: (Fig. 6a-6c) should be (Fig. 6a-6c, Fig. 7a-7c).

Corrected. Thank you.

---

## Author Comment (AC2)

**Reviewer #3**

This article provides an excellent work on local to large scale interaction of meteorological parameters for precipitation. It well defines the mesoscale and isolated convective systems over southeastern Texas region. There are some minor revisions needed before acceptance of the manuscript.

5    We thank the reviewer for their thoughtful and constructive comments and suggestions, which has substantially improved the quality of the manuscript. We have addressed all the reviewer's concerns and revised the manuscript accordingly. Our point-by-point responses are in blue and the modifications to the manuscript are quoted in green.

1. In introduction, provide more detailed description of local scale features and precipitation over Texas region.

The following paragraph has been added to the introduction:

10    *Southeastern Texas is characterized by high annual precipitation, attributable to its abundant moisture from the Gulf of Mexico, with significant seasonal variation (Statkewicz et al., 2021). This area is prone to intense summer convection storms, driven by sea breeze and daytime surface heating, as well as the impacts of tropical cyclones during the hurricane season (Caicedo et al., 2019; Darby, 2005). The winter season is typically marked by rainfall from cold fronts, while spring can see severe weather events like thunderstorms and occasionally tornadoes (Prat and Nelson, 2014). Nocturnal thunderstorms are*
15    *common due to the warm, moist air transported by the GPLLJ from the Gulf (Berg et al., 2015; Day et al., 2010). Meanwhile, urban development in Houston metropolitan exacerbates flooding risks by reducing natural land absorption capacity (Van Oldenborgh et al., 2017; Chang et al., 2007; Burian and Shepherd, 2005). Precipitation patterns are influenced by both large-scale and local factors such as urbanization and sea breezes.*

2. Line number 44: "four large-scale meteorological patterns…...". What are these four? Please explain.

20    The four large-scale meteorological patterns identified in Wang et al. (2022) are pre-trough, post-trough, anticyclone, and transitional regimes. It has been clarified in the revised manuscript.

3. What is the reason of taking 2004 – 2017 year for analysis?

The convection dataset spans from 2004 to 2017, constrained by the source data availability during the time of dataset creation, which includes Stage IV precipitation data and GridRad (3-D Gridded NEXRAD WSR-88D Radar Data).
25    Although a newer version of GridRad is extended to 2021, it only begins in 2008. We consider the 14-year period is sufficiently long for robust analysis.

4. In Figure 2 and 3, are those days are only precipitation days or all days in each season?

In both figures, the upper panels (a-d) are mean of all hours in each season, while the lower panels are anomalies for the hours only with convective precipitation in the study area.

30   5. In Figure 4 and 5, provide latitude values along y-axis for a, f, & k subplots, and longitude values along x-axis for q, r, s, & t subplots.

Modified as based on reviewer's suggestion.

6. How have you calculated pre-front, front, and post-front? Is it based on days or hours? Is it depending on any meteorological conditions?

35 The hourly input vectors ($u$, $v$, and $q$) are clustered into four groups using the SOM technique. The hours assigned to the same SOM type are averaged to produce the four LSMPs, namely pre-front, front, post-front, and anticyclone. In addition to the variables used for SOM clustering, we utilized vertical velocity and surface temperature to help interpret the dynamical processes associated with each LSMP.

7. In Figure 6 and 7, it is better to remove wind vector text near to color bar, as it is confusing. What is the difference
40 between red and black wind vectors?

The "wind vector" text has been removed. We have denoted vectors statistically significant at the 5% level with purple, as determined by the student's t-test (please refer to the caption of Figure 6 for details).

---

## Author Comment (AC3)

**Reviewer #2**

Review of "Tracking precipita3on features and associated large-scale environments over southeastern Texas" by Y. Liu et al.

General Comments:

This paper examines the characterizes of convection systems, i.e., mesoscale convective systems (MCSs) and isolated deep convec3on (IDC), and their associated large-scale meteorological patterns (LSMPs) over southeastern Texas in different seasons from 2004 to 2017. The tracks of MCSs and IDC in the region and the diurnal cycle of IDC in the Houston metropolitan area are also explored. While the study holds great potential for enhancing the understanding of the spatiotemporal features of convective systems in southeastern Texas, the paper would be more convincing with
improved clarity in methodology and analysis.

We thank the reviewer for their thoughtful and constructive comments and suggestions, which has substantially improved the quality of the manuscript. We have addressed all the reviewer's concerns and revised the manuscript accordingly. Our point-by-point responses are in blue and the modifications to the manuscript are quoted in green.

Specific comments:

1. I think the motivation of the study is not quite clear (lines 42-47). In addition to extending previous studies in summer to other seasons, it would be more interesting to highlight what's new in this study and what specific research questions this paper tries to address. It would also be helpful to briefly review previous findings about MCSs and IDC in this region.

We have added the following paragraphs to the introduction (Lines 42-65) to (1) briefly review the previous findings
about MCSs and IDC in this region, (2) highlight the novel aspects of this study, and (3) summarize the science questions addressed.

*Southeastern Texas is characterized by high annual precipitation, attributable to its abundant moisture from the Gulf of Mexico, with significant seasonal variation (Statkewicz et al., 2021). This area is prone to intense summer convection storms, driven by sea breeze and daytime surface heating, as well as the impacts of tropical cyclones during the hurricane season*
*(Caicedo et al., 2019; Darby, 2005). The winter season is typically marked by rainfall from cold fronts, while spring can see severe weather events like thunderstorms and occasionally tornadoes (Prat and Nelson, 2014). Nocturnal thunderstorms are common due to the warm, moist air transported by the GPLLJ from the Gulf (Berg et al., 2015; Day et al., 2010). Meanwhile, urban development in Houston metropolitan exacerbates flooding risks by reducing natural land absorption capacity (Van Oldenborgh et al., 2017; Chang et al., 2007; Burian and Shepherd, 2005). Precipitation patterns are influenced by both*
*large-scale and local factors such as urbanization and sea breezes. The primary aim of this study is to isolate the impact of large-scale circulations on various convection systems, distinguishing it from other contributing factors, and to pinpoint the large-scale conditions that promote the development of sea-breeze circulations.*

*To understand the role of large-scale and local factors in convection initiation and development, Wang et al. (2022) isolated the influence of large-scale meteorology from microphysical and mesoscale influences, with the focus on summer climate*
*over southeastern Texas. They identified four large-scale meteorological patterns (LSMPs), namely pre-trough regime, post-trough, anticyclone, and transitional regimes, characterized by the location and strength of the Bermuda High and the GPLLJ. Both the Bermuda High and GPLLJ are seasonally varying systems, which exert different impacts on different types of convection.*

*In this study, we extend the analysis from summer to all four seasons during 2004-2017, and separate LSMPs associated*
*with MCSs and IDC. By using a convection dataset (Li et al. 2021a), we investigate the track trajectories under various LSMPs since the track properties are influenced by the land surface and meteorological conditions along their pathways. Moreover, we apply the cell tracking method (Feng et al., 2022) to the Next-Generation Radar (NEXRAD) system to obtain fine-scale (500 m) track features (e.g., initiation location and timing) over Houston. We then assess the variation of these fine-scale features with LSMPs. Specifically, we address the following questions in this study: (1) How much do various*
*precipitation types contribute to the seasonal rainfall in southeastern Texas? (2) How do different LSMPs affect various types of convective precipitation across different seasons? (3) How do LSMPs influence the trajectories and associated MCSs and IDC's statistical properties?*

2. The 4-km convection dataset is the key dataset used in this study (section 2.1). It would be helpful and informative to briefly explain how MCSs, IDC, and non-convective systems are identified in the dataset and discuss any
uncertainties related to the methodology. Adding information about how to separate tropical cyclone-induced precipitation from total precipitation in the dataset (lines 70-73) would be helpful, too.

The following paragraph has been added for the definition of MCS and IDC, in Lines 86-93.

*A MCS event is defined if it satisfies the following criteria: (1) there is at least one pixel of cold cloud core during the whole life cycle of the track, (2) CCS areas associated with the track surpass 60 000 km$^2$ for more than 6 continuous hours, and (3)*
*PF major axis length exceeding 100 km and intense convective cell areas of at least 16 km$^2$ exist for more than 5 consecutive hours. For the non-MCS tracks, an IDC event is defined with the following two criteria: (1) a CCS with at least 64 km2 is detected, and there is (2) at least 1 h during the life cycle of the track when PF and CCF are present (PF and CCF major axis lengths $\geq 4$ km). Li et al. (2021a) discussed the impact of chosen threshold in separating MCS and long-lasting IDC, and concluded that the current criteria well capture the spatial distribution and essential characteristics of MCS/IDC*
*precipitation.*

The following text are added for the separation of tropical cyclone-induced precipitation.

*Following Li et al. (2021b), this study further separates tropical cyclones (TCs) and associated precipitation from the convection dataset using the historical and most recent TCs obtained from the IBTrACS (International Best Track Archive for Climate Stewardship) version 4.0 dataset (Knapp et al., 2010), using the following approach. (a) MCS/IDC tracks with*
*their cold clouds overlapping with a TC at any time during its lifecycle are considered parts of the TC. (b) Any non-MCS/IDC clouds overlapping with a TC are included as parts of the TC. (c) All grid cells overlapping with a TC at any given time are parts of the TC at that specific time. Finally, we can distinguish the precipitation type of each precipitating grid cell based on the combined data set: MCS, IDC, TC, or non-convective system.*

3. Similarly, it is preferred to add some discussion about the potential uncertainties associated with the KHGX
reflectivity and the method to identify convective cells, if any.

The definition of convective grid point is modified from Steiner et al., 1995. The threshold adjustment aims to better identify individual deep convective cells from the background, an approach that was effectively applied in the CACTI study (Feng et al., 2022). Another modification from Steiner et al., 1995 is the $Z_{bkg}$-dependent radius step function to define boundary of convective cells. While this function influences the area of the convective cell, we only use KHGX radar
data to assess the location and time of convection initiation. We have added the following sentence to the revision in Lines 125-126:

*While the step function influences the area of the convection cell, we only use KHGX radar to study the location and time of convection initiation.*

4. The usage of 'the initial hour' (line 120) is not very well explained. Is it exactly one hour (or a few hours) ahead of the initialization of convective systems or just the first hour of the convection? If it is just an hour before or at the time of convection, is this time frame too short or sufficient to capture the large-scale features? Have you examined a few hours before the occurrence of MCSs and IDC? Are the results similar or different?

We use the first hour of the track defined in the convection dataset. We've done a similar analysis in the past (Song et al., 2019) with MCS LSMP. The argument is that the large-scale environments (>1000 km scale) do not change significantly within 3–6 hours. Using the initiation hour is sufficient to capture the large-scale circulation patterns using SOM.

5. In section 3.2, self-organizing maps (SOM)-identified large-scale meteorological patterns (LSMPs) are categorized into three frontal-system patterns and an anticyclone pattern for MCSs and two frontal-related and two anticyclone-related patterns for IDC, but without explanation about why and how these categories are defined. Taking Fig. 7h as an example, the circulation in the black box (purple arrows) shows a somewhat cyclonic pattern, but it is labeled as a type of 'Anticyclone' pattern.

The SOM analysis utilizes $u$, $v$, and $q$ values at each hour to construct the input vector. Hours that exhibit similar patterns in their input vectors are grouped under the same SOM cluster. The meteorological patterns for each SOM cluster are determined by a composite average of the hours within that category. Each cluster is named according to the characteristics of the large-scale circulation and those from previous studies (e.g., Wang et al., 2022). In the revised manuscript, we have overlaid the anomalous 500-hPa geopotential height onto Figures 6 and 7. Specifically, in Figure 7 (referenced below), both panels 7d and 7h feature an anomalous high at 500 hPa, inducing an anticyclone feature over the continent, although the magnitudes of these anomalous highs are considerably smaller in the summer than in the spring.

[Figure]

Figure 7: Anomalous wind vector (m s⁻¹) and specific humidity (g kg⁻¹) at 925 hPa for spring (a-d) and summer (e-h) associated with IDC initiation. Contours are the 500-hPa geopotential height (gpm) anormalies. Numbers in the parentheses indicate the frequency of each LSMP. Wind vector and specific humidity below terrain are masked out. Significant vectors at the 5% level according to the student's t-test are shown in purple. Only significant specific humidity anomarlies are shown. The black box indicates the region for SOM analysis.

The following sentences have been added to the method section to clarify how the clusters are named, in Lines 164-166:

*For each SOM cluster, the meteorological patterns are determined by a composite average of the hours within that cluster. Each cluster is named according to the characteristics of the large-scale circulation and those from previous studies (e.g., Wang et al., 2022).*

6. It's not very clear why the GPLLJ 'weakens' during summer (line 144). Does this refer to a specific section of the jet?
Previous studies showed that the GPLLJ peaks in June-July (e.g., Weaver and Nigam 2008). The meridional winds over the Great Plains region also appear stronger in JJA than in MAM in Fig. 2a-b.

We agree with the reviewer that the GPLLJ peaks in summer. Thanks for pointing this out. We have removed this statement in the revision.

7. Specific humidity at 925 hPa is used in the analysis in different places of the paper (e.g., Figs. 2, 6, 7). While this is
the level close to the core of the GPLLJ and probably well reflects low-level moisture transport, technically, precipitation is more directly related to vertically integrated moisture convergence (and also surface evapotranspiration). I wonder if you have tried vertically integrated moisture flux or moisture convergence in the analysis and if you get similar results.

Per reviewer's suggestion, we have replaced the specific humidity at 925 hPa with the vertically integrated moisture
convergence and redone the SOM analysis. The results are consistent with the one using q925. In the following figures (Fig. R1 and R2), we plot the same variables as Figure 6 and Figure 7 (winds and specific humidity at 925 hPa for spring and summer for MCS and IDC, respectively), but with the SOM clusters based on the integrated moisture.

[Figure]

[Figure]

**Figure R2: The same as Figure 7 in the manuscript, but with SOM clusters based on the vertically integrated moisture.**

8. In Section 3.4, I'm curious if there are also a lot of IDC events during the fall as shown in Table 2. It would be nice to have results (like Fig. 11) from other seasons in the supplementary materials.

In the revision, we have modified Figure 11 to include results from fall. The number of IDC initiated in the radar coverage in spring and winter are much fewer than summer and fall. In fall, the spatial distribution of IDC initiation is generally consistent with that in summer. There are slightly more IDC initiated over ocean under anticyclone conditions in fall.

     The following texts are added to the revision, in Lines 358-364:

*A significant number of IDC also occur in fall. The spatial distribution of IDC initiation locations in fall generally aligns with those in summer (Fig. 11e-h). However, fall is characterized by a front LSMP rather than the weak-anticyclone LSMP identified in summer. The front LSMP favors IDC initiation over land, although this LSMP contributes minimally to the total number of IDC. Under anticyclonic conditions, the occurrences of IDC initiation over the ocean slightly increases in fall compared to those in summer, due to the relatively weaker sea-breeze circulation in fall.*

[Figure]

**Figure 11. Number of IDC initialed during each LSMPs in summer (a-d) and fall (e-h). The purple dot indicates the location of the KHGX site. The solid and dashed circles indicate radii of 110 km, 100 km, and 50 km.**

Technical corrections:

1. Fig. 2, it's better to mask out wind vectors and specific humidity over the mountainous region where the surface is above the 925 hPa level. In the caption, it says "precipitation" but the text indicates it's convective precipitation (lines 120-121). Can you please clarify?

Wind vectors and specific humidity over the mountainous regions are masked out in Figure 2, as well as in Figure 6
and 7. Figure 2 shows the convective precipitation, which has been updated in the revision and is referenced as follow.

[Figure]

**Figure 2: Mean (a-d) and anomalous (e-h) wind vector (m s⁻¹) and specific humidity (g kg⁻¹) at 925 hPa for each season. Wind vectors and specific humidity below the terrain are masked out. The anomaly is the difference between the hours with MCS/IDC assocaited precipitation in southeastern Texas and the climatology. The southeastern Texas area is marked by the purple box (28°N – 32°N, 97°W – 93°W).**

2. Figs. 6-7, can you please add the frequency (%) to each LSMP pattern to the figures?

Frequencies have been added to Figures 6 and 7. The correspond caption has been updated as well.

3. Fig. 12, is this for summer or all the seasons?

Fig. 12 is just for summer. It has been clarified in the revision.

4. Lines 319-320, which figure do you refer to?

This sentence refers to Figure 6e and g. It has been added to the revision.